# Lifetime Comparisons of Organic Light-Emitting Diodes Fabricated by Solution and Evaporation Processes

**DOI:** 10.3390/mi14020278

**Published:** 2023-01-21

**Authors:** Kuo-Chun Huang, Jeng-Yue Chen, Yen-Hua Lin, Fuh-Shyang Juang, Yu-Sheng Tsai

**Affiliations:** 1Department of Electro-Optical Engineering, National Formosa University, Huwei, Yunlin 632-08, Taiwan; 2Department of Electrical Engineering, National Formosa University, Huwei, Yunlin 632-08, Taiwan

**Keywords:** solution process, blue fluorescent material, current balance of electron–hole, lifetime extension

## Abstract

In this paper, a blue fluorescent organic light-emitting diode (OLED) with a 1 cm^2^ emitting area was fabricated by a solution process. The ITO/spin MADN:13% UBD-07/TPBi/Al was used as the basic structure in which to add a hole-injection layer PEDOT:PSS and an electron-injection layer LiF, respectively. The device structure was optimized to obtain a longer lifetime. Firstly, the TPBi, which is an electron transport layer and a hole-blocking layer, was added to the structure to increase the electron transport rate. When the TPBi thickness was increased to 20 nm, the luminance was 221 cd/m^2^, and the efficiency was 0.52 cd/A at a voltage of 8 V. Since the addition of the hole-injection layer (HIL) increased the hole current but did not increase the electron current, the electron transport layer (ETL) Alq_3_ with the lowest unoccupied molecular orbital (LUMO) was added as stepped ETL to help the TPBi transport more electron current into the emitting layer. When the thickness of the TPBi/Alq_3_ was 10 nm/15 nm, the luminance reached 862 cd/m^2^, the efficiency was 1.29 cd/A, and the lifetime increased to 252 min. Subsequently, a hole-injection layer PEDOT:PSS with a thickness of 55 nm was added to make the ITO surface flatter and to reduce the probability of a short circuit caused by the spike effect. At this time, the luminance of 311 cd/m^2^, the efficiency of 0.64 cd/A, and the lifetime of 121 min were obtained. Following this, the thickness of the emitting layer was doubled to increase the recombination probability of the electrons and the holes. When the thickness of the emitting layer was 90 nm, and the thermal evaporation method was used, the efficiency was 3.23 cd/A at a voltage of 8V, and the lifetime was improved to 482 min. Furthermore, when the thickness of the hole-injection layer PEDOT:PSS was increased to 220 nm, the efficiency increased to 3.86 cd/A, and the lifetime was increased to 529 min. An infrared thermal image camera was employed to detect the temperature variation of the blue OLEDs. After the current was gradually increased, it was found that the heat accumulation of the device became more and more significant. When the driving current reached 50 mA, the device burnt out. It was found that the maximum temperature that the OLED device could withstand was approximately 58.83 degrees C at a current of 36.36 mA.

## 1. Introduction

Significant efforts are focused on improving the efficiency, stability, and color tunability of organic light-emitting diodes (OLEDs) in order to make them more suitable for future displays and ambient lighting [1]. Currently, white organic light-emitting diodes (WOLEDs) have been commercialized as general solid-state lighting. Unlike inorganic light-emitting diodes (LEDs), OLEDs offer high flexibility in spectral customization because of the wide variety of organic emitters, from red to violet. WOLEDs can be easily realized by stacking two or more emitters of different colors over a large area [2]. OLEDs have the characteristics of self-luminescence and also have the advantages of thinness, wide viewing angle, full color, low power consumption, and flexibility. In 2017, a leading mobile phone company, APPLE, launched the iPhone X. The screen used in the iPhone X was the Active Matrix Organic Light Emitting Diode (AMOLED), the use of which caused an increase in OLED-related research topics. Although OLEDs are superior to LCD in characteristics, thermal evaporation vacuum deposition technology faces many disadvantages, such as low material utilization rate and high energy consumption [3]. Within the OLED category, blue organic light-emitting diodes (blue OLEDs) have attracted attention due to their applications in full-color flat-panel displays and lighting. The blue emitter acts as one of the three primary colors or as a color converter light source to obtain green and red emissions from the fluorescent material. Conventional blue OLEDs have lower efficiency and shorter operating lifetimes compared to green and red OLEDs [4]. Once the blue light is attenuated, it causes the phenomenon of color shift to occur to the whole screen. This shortcoming forces each technology factory to develop a solution. In addition to the short lifetime of OLEDs, the organic materials used in OLEDs are also prone to material deterioration caused by water vapor and oxygen intrusion. As such, the question of how to prolong the lifetime of OLED devices is a very important research topic.

In this study, the electron transport layer (ETL) TPBi was added to increase the electron transport rate. A second ETL Alq_3_ was added as a stepped ETL to help the TPBi to transport more electron current into the emitting layer. The thickness of the TPBi/Alq_3_ was studied. The optimum thickness of the hole-injection layer PEDOT:PSS was studied, which was employed to make the ITO surface flatter. The emitting layers deposited with the spin-coating and the thermal evaporation were compared in order to extend the OLED lifetime. An infrared thermal image of the OLED emission was measured to study the temperature variation of the blue OLEDs’ emissions.

## 2. Experimental Section

In order to make the OLED device emit light, it was necessary to define the cathode and the anode areas, respectively, and make them mutually non-conductive. At this time, it was necessary to use AutoCAD software to draw the designed pattern and effective light-emitting area. Subsequently, indium tin oxide (ITO) glass was engraved with the laser engraving system to define the patterns.

OLEDs are easily affected by a poor environment, such that even tiny dust or dirt particles on the substrate will cause internal cracking of the device, which would reduce the luminance efficiency and seriously reduce the lifetime of the device. In order to ensure that the substrate was in a pollution-free state, the ITO substrate was cleaned and surface-treated. The ITO substrates were washed in acetone, isopropanol, and de-ionized water sequentially for 20 min using an ultrasonic vibration machine. After the ITO substrate was blown dry with a nitrogen (N_2_) gun, it was sent to the oven for continuous baking at 80 °C for 1 h to avoid cracking of organic materials due to residual moisture left on the substrate. After drying, it was sent to the UV ozone surface treatment system. 

Next, the organic layers were spin-coated using the solution processes. The emitting material was prepared first. The solution composition was the MADN host material: UBD-07 guest material = 8.7 mg: 1.3 mg with 1 c.c. chlorobenzene as solvent, where MADN is 2-methyl-9,10-bis(naphthalen-2-yl)anthracene and UBD-07 is 4,4′-Bis(9-ethyl-3-carbazovinylene)-1,1′-biphenyl. The solution was then placed on a hot plate at 50 °C and stirred continuously with a magnet at 550 rpm for at least 6 h to ensure uniform dissolution. One c.c. hole-injection layer material PEDOT:PSS (Poly(3,4-ethylenedioxythiophene) polystyrene sulfonate) was filtered through a 0.45 μm filter to remove excessively large particles in the solution, then dropped onto the substrate, and spin-coated at 2500 rpm for 30 s. After the coating was completed, the substrate was immediately transferred onto the hot plate and baked at 110 °C for 20 min.

After the PEDOT:PSS was dried, the prepared emitting material was uniformly spin-coated on PEDOT:PSS at 2500 rpm for 30 s. After the coating was completed, it was immediately transferred onto a hot plate for drying at 110 °C for 20 min. In this study, after the HIL and the emitting layer were spin-coated, the electron transport layer (ETL) was deposited by vacuum thermal evaporation. The reason for this was that if the solution process was continuously employed, there was a high probability that the ETL (that also works as a hole-blocking layer) and the emitting layers would dissolve into each other, and the internal structure of the OLEDs would lose their original operating function. In this study, a glass cover plate and UV glue were used to seal and encapsulate the OLED devices in a nitrogen glove box so as to avoid the penetration of external water and oxygen into the devices and damage to the organic materials. The full chemical names and the molecular structures of materials used in this work are shown in Figure 1.

In order to measure the electro-optical characteristics of the OLED, the fabricated device was placed on a fixture, then a Photo Research PR 650 (a spectra colorimeter, Photo Research Inc., Chatsworth, CA, USA) was aligned with the emitting area, and a Keithley 2400 (a power supply, Keithley, Solon, OH, USA) was used to provide the bias to the OLED. The current–voltage–luminance (I–V–L) characteristics and the emission spectrum were measured. For the device lifetime measurement, a self-designed measurement system (with Raspberry Pi and Arduino module) was employed to automatically record the luminance data, and in this way, the improvement of the device lifetime was studied. For all lifetime measurements, a constant current source was employed to provide 20 mA/cm^2^ to drive the packaged blue OLEDs in a room-temperature environment. An infrared thermal image camera (NEC Thermo Tracer TH7716, NEC Avo Infrared Technologies Co., Ltd, Shinagawa-ku, Tokyo) was employed to detect the temperature variation of blue OLEDs.

## 3. Results and Discussion

The emitting layer in this study was compared using two deposition methods. The first method was deposition by a solution process. The host material of MADN and the blue guest material UBD-07 were mixed and dissolved in a chlorobenzene solvent. The ratio was MADN with 13% UBD-07, which was spin-coated on an ITO substrate. The basic structure was ITO/PEDOT:PSS/MADN:13% UBD-07 (45 nm)/LiF/Al (150 nm). The second method was deposition by vacuum thermal evaporation.

In the basic structure, TPBi (2,2′,2″-(1,3,5-Benzinetriyl)-tris(1-phenyl-1-H-benzimidazole)) was added as both an electron transport layer (ETL) and a hole-blocking layer (HBL). The device parameters are shown in Table 1. The electron transport properties of the TPBi were employed to increase the effect of the electrons being injected into the emitting layer from the cathode. Moreover, through the HOMO energy barrier between the TPBi and emitting layer, as shown in Figure 2, the holes were confined to the emitting layer, which promoted the recombination of the electrons and the holes to emit light [5].

In addition to employing TPBi as the ETL, PEDOT:PSS was also spin-coated on the ITO as the hole-injection layer (HIL) (device A). The energy band structure is shown in Figure 1a, and the parameters of each layer of device A are shown in Table 1. The purpose of adding PEDOT:PSS was to improve the surface flatness of the ITO. By covering the surface of the ITO with PEDOT:PSS in the form of an aqueous solution, the short circuit caused by the spike effect could be reduced. Moreover, through the work function of PEDOT:PSS (−5.2 eV) to reduce the energy barrier between the work function of the ITO and the HOMO of the emitting layer (EML), it was easier for holes to be injected from the ITO into the emitting layer. The accumulation of holes in the ITO/EML interface could be avoided, which also reduced the heat generation and prevented the devices from being easily burned out [6].

In this study, in addition to using TPBi as the ETL and HBL, Alq_3_ (Tris(8-hydroxyquinoline)aluminum-(III)) was also added as the electron transport layer (device B). The larger lowest unoccupied molecular orbital (LUMO) energy level of Alq_3_ (lower energy position) was utilized to help the electrons inject more smoothly from the cathode to the TPBi and then to the emitting layer, resulting in an increase in current density.

However, the total thickness of the hole-blocking layer TPBi and the electron transport layer Alq_3_ should not be too thick in order to prevent the overall resistance of the device from being too high and affecting its optoelectronic properties. We used a thickness of TPBi 20 nm in device A and adjusted the thickness of the TPBi/ Alq_3_ to 10/15 nm (device B) in order to help more electrons be transported to the emitting layer by increasing the thickness of Alq_3_. The parameters of device B are shown in Table 1, and the energy band structure is shown in Figure 2b. The electronic mobility of TPBi and Alq_3_ was 3.3–8 × 10^−5^ cm^2^/V·s [7,8] and 9.4 × 10^−6^–1.1 × 10^−5^ cm^2^/V·s [9,10], respectively. Although the electron mobility of TPBi was higher than that of Alq3, the energy difference between the LUMO of TPBi (−2.7 eV) and the work function of Al (−4.2 eV) was too large, which made electron injection difficult.

The LUMO energy level of Alq_3_ was −3.2 eV, which was between the LUMO of TPBi and the work function of Al, as shown in Figure 2b. Therefore, the electron-injection energy barrier was reduced so that electrons could be injected from the cathode to the TPBi more smoothly, reducing both the accumulation of electrons at the Al/TPBi interface and the thermal effect. Consequently, the lifetime of the device was increased [5].

In the solution process, the emitting layer was spin-coated twice to increase the thickness of the emitting layer from 45 nm (device B) to approximately 90 nm (device C). The thickness of the EML in the device was increased in order to increase the probability of electrons and holes recombining in the emitting layer. However, the solution process can easily cause damage to the film during repeated coating. The film morphology for the EML of devices B, C, and D, respectively, were measured with an atomic force microscope (AFM), as shown and compared in Figure 3. The root mean square deviation values (R_q_) of surface roughness are also shown in Figure 3. The results showed that the EML surface with two spin-coatings decreased in quality. Therefore, it was proposed to change the fabrication method of the EML to vacuum thermal evaporation (device D) instead of the solution process. The optoelectronic properties of the two are compared. The structural parameters of the devices are shown in Table 1.

The emitting layer of device C was deposited twice by spin-coating. The second spin-coating may have caused the first spin-coating layer to dissolve, resulting in the poor quality of the emitting layer and uneven thickness of the film. This would lead to an uneven current distribution of the devices, which would increase the decay of the device lifetime; therefore, the lifetime of device C is worse than that of device B, as shown in Figure 4a. Figure 4b shows the fitting curves with the formula of Equation (1) [11]. The fitting of Figure 4b is executed by the User Defined Fitting Function (Equation (1)) in the Origin software. For device B: *τ* = 619.4, *β* = 0.4.
(1)$$L\left(t\right)/L_{0}=\exp[-\left(t/\tau {)}^{\beta }\right]$$

For device D, the emitting layer was changed to a vacuum thermal evaporation process, and the thickness was maintained at 90 nm. The vacuum evaporation (@8 × 10^−6^ torr) had better film-forming properties than spin-coating, and there were no mutual dissolution or uneven thickness issues resulting from the spin-coating. The AFM film morphology of device D is shown in Figure 3. The surface roughness *R_q_* of device D was 2.6 nm. Its luminance and efficiency were greatly improved, as shown in Figure 5. At 9V, the current density of the device with the EML deposited by the vacuum thermal evaporation compared to the solution processes was increased from 27.4 mA/cm^2^ to 41.0 mA/cm^2^; the luminance from 156.4 to 1339 cd/m; the efficiency from 0.57 to 3.27 cd/A; and the lifetime also improved from 165 min to 482 min.

In this study, in order to improve the lifetime and stability of OLEDs, an attempt was made to increase the thickness of the hole-injection layer (PEDOT:PSS) to observe its influence on the optoelectronic properties. The structural parameters of device E are shown in Table 1.

From the luminance–voltage (L–V) characteristic curves in Figure 6, it can be seen that when the thickness of PEDOT:PSS increased from 55 nm (device D) to 220 nm (device E), the hole-injection resistance at the anode terminal increased slightly, resulting in a decrease in the hole current. The number of electrons injected at the cathode terminal was unchanged, so the luminance was not affected (the luminance was not increased). It is inferred that increasing the thickness of PEDOT:PSS can reduce the number of holes entering the emitting layer, making the electron–holes recombine more effectively, thus improving the efficiency and lifetime of OLEDs. If the PEDOT:PSS thickness increased, the e–h recombination and light-emitting position would be far away from the anode (ITO), so the exciton quenching effect may be reduced, as discussed in Ref. [12]. From the literature, Reference [6] stated: “In samples that contained a PPV film (emitting layer) on top of the ITO/PEDOT:PSS film, a rather sharp decrease in the indium concentration was observed at a depth corresponding to the PEDOT:PSS/PPV interface. This implies that the indium containing etch products do not diffuse into the PPV film, but are trapped in the PEDOT:PSS layer” [6]. As such, a thick PEDOT:PSS layer is necessary between the ITO and the emitting layer. When the thickness of the HIL PEDOT:PSS increased, the resistance in the anode side increased to reduce the accumulation of holes; thereby, the lifetime of the OLED was prolonged from 482 to 529 min. At 8 V, the luminance efficiency was also improved from 3.23 to 3.86 cd/A.

Figure 7a shows the total comparison of the lifetime of each device. The EML in device A was thinner due to the solution process, and there was no additional Alq_3_ ETL layer, so the lifetime was the lowest. After adding 15 nm Alq_3_ to device B, the energy level at the cathode side was buffered, which made it easier for electrons to be injected into the EML, thereby reducing the problem of electron accumulation and prolonging the lifetime slightly. The lifetime was increased from 121 to 252 min. The thickness of the EML in device D was increased from 45 to 90 nm, which increased the number of electron–hole recombinations in the EML. Additionally, its deposition method changed from the solution process to the thermal evaporation process, which increased the current efficiency and lifetime. From the literature, Reference [13] stated: “the intermolecular distances were larger in the spin-coated film than in the evaporated one indicating that the molecular packing is denser in the latter case.” In that study, the lifetime was increased from 252 to 482 min.

After the PEDOT:PSS HIL thickness in device E was increased from 55 to 220 nm, the number of holes injected into the EML was reduced, the electron–hole current imbalance was reduced, and the luminance current efficiency was improved, as shown in Figure 8. Finally, the lifetime was increased from 482 to 529 min, as shown in Figure 7a. Figure 7a shows the experimental data of lifetime. Figure 7b shows the fitting curves with the formula of Equation (1) [11]. For curve A: *τ* = 222.46, *β* = 0.41; B: *τ* = 619.4, *β* = 0.4; curve D: *τ* = 729, *β* = 0.77; E: *τ* = 844.65, *β* = 0.77.

## 4. Study on the Thermal Effect of Blue OLED

A constant current source was employed to provide 5 mA to drive the packaged blue OLED, and a thermal infrared camera was used to measure the surface temperature of the device, as shown in Figure 9. Reference [14] shows a similar study. The surface temperature was measured every 40 min for two hours. It was found that the surface temperature was about 30.3 °C, and there was no obvious change during the 120 min, as shown in Figure 10. In order to understand the relationship between the lifetime decline and the surface temperature of the blue OLED, the driving current was further increased at intervals of 10 mA, and the change in the surface temperature was observed, as shown in Figure 11. As the current increased, the surface temperature also increased, and it can be seen that the accumulation of thermal energy was becoming more and more significant. It can be seen from the temperature distribution diagram of the center of the OLED in Figure 11 that the temperature of the OLED increased as the current increased from 5 to 40 mA and its temperature gradually increased from 30.1 °C to 57.2 °C. When the current continued to increase to more than 40 mA, the temperature of the OLED exceeded the range that the organic material could withstand; subsequently, it burned out, and the temperature decreased after the burnout. The relationship between the OLED center temperature and the current is shown in Figure 12. Curve fitting is performed on the device temperature and current data, as shown by the red dashed line in Figure 12. It was found that the maximum temperature that the OLED device could withstand was approximately 58.83 °C at a current of 36.36 mA.

## 5. Summary

In this study, ITO/MADN:13% UBD-07/TPBi/LiF/Al was employed as the basic structure. After adding 15 nm Alq_3_ to the ETL, the energy level at the side was buffered, which made it easier for electrons to be injected into the EML, and as such, the electron accumulation problem was reduced. The OLED lifetime was slightly extended from 121 to 252 min. In device D, the thickness of the EML was increased from 45 to 90 nm, which increased the number of electron–hole recombinations in the EML. The deposition method of the EML was changed from a solution process to a vacuum thermal evaporation process, and the luminous efficiency and the lifetime were increased. The lifetime was increased from 252 to 482 min. In device E, the PEODT:PSS thickness was increased from 55 to 220 nm, the number of holes injected into the EML was reduced, and the electron–hole number imbalance was reduced, which resulted in the improvement of the current efficiency and lifetime. Finally, the lifetime of the OLED was increased from 482 to 529 min.

The temperature variation on the surface of the blue OLED was measured using a thermal infrared camera. The temperature variation of the OLED driven at a constant current of 5 mA was extremely small. When the current was gradually increased, it was found that the heat accumulation became more and more significant. Additionally, when the current was increased to 40 mA, the entire device was burned out. It can be deduced that the maximum temperature that the OLED in this study can withstand is approximately 58.83 °C when the driving current is 36.36 mA.

## Figures and Tables

**Figure 1 micromachines-14-00278-f001:**
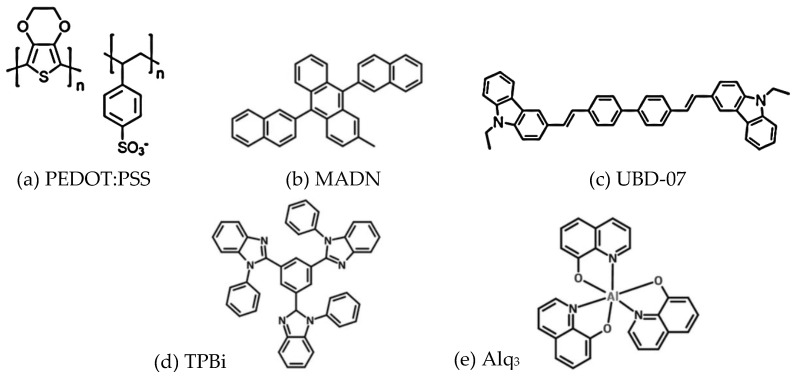
The molecular structures of materials used in this work.

**Figure 2 micromachines-14-00278-f002:**
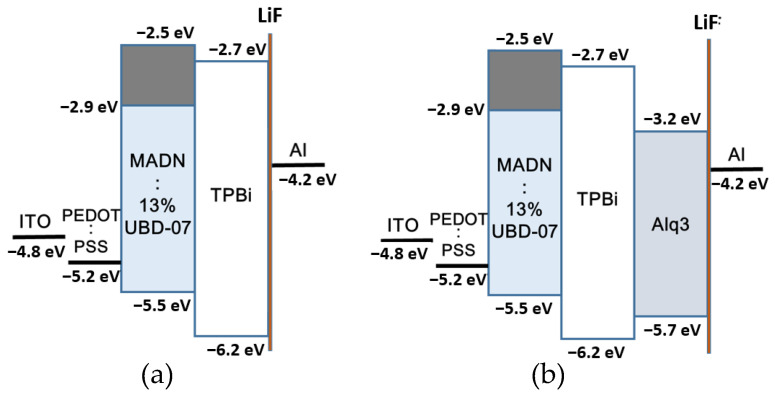
Band diagram of (**a**) adding a HIL PEDOT:PSS, (**b**) adding an ETL Alq_3_, where light blue region is the energy band of dopant.

**Figure 3 micromachines-14-00278-f003:**
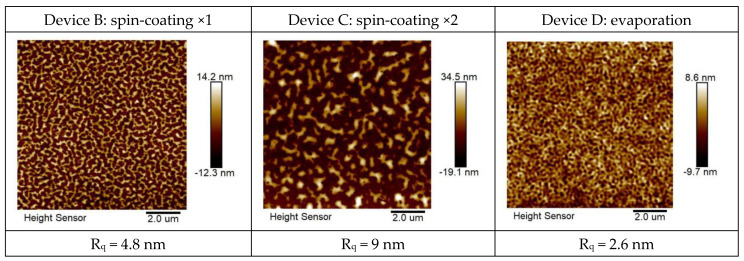
AFM film morphology for the EML of devices B, C, and D, respectively.

**Figure 4 micromachines-14-00278-f004:**
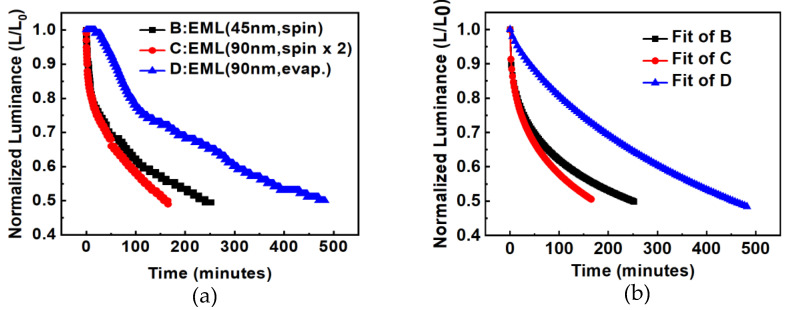
Lifetime measurement of OLEDs with different EML deposition methods and thicknesses, (**a**) original curves, (**b**) fitted curves.

**Figure 5 micromachines-14-00278-f005:**
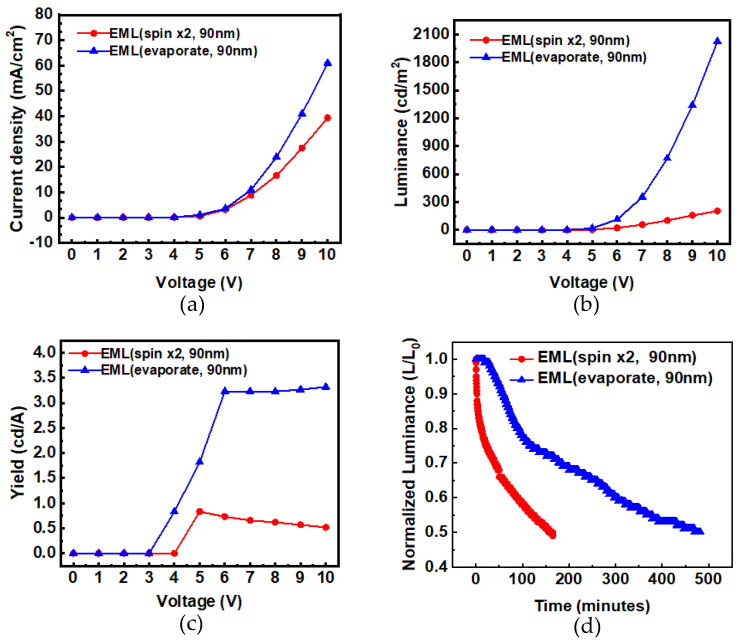
Comparisons of (**a**) I−V, (**b**) L−V, (**c**) η−V, and (**d**) lifetime characteristics of spin-coated EML compared with vacuum-evaporated.

**Figure 6 micromachines-14-00278-f006:**
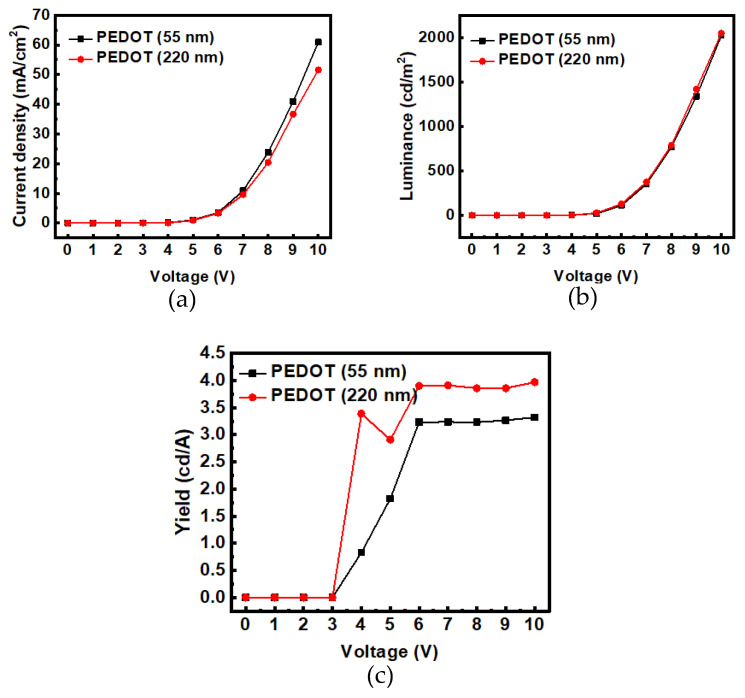
The comparisons of characteristic curves of thick HIL with thinner HIL: (**a**) I−V, (**b**) L−V, and (**c**) η−V.

**Figure 7 micromachines-14-00278-f007:**
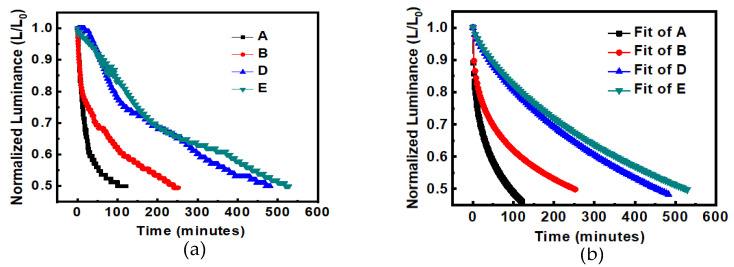
Comparisons of device lifetime (**a**) experimental data, (**b**) fit curves with the formula.

**Figure 8 micromachines-14-00278-f008:**
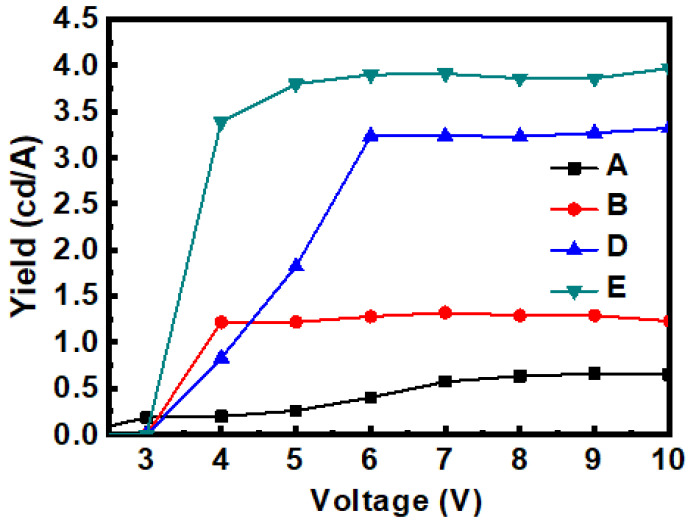
Current efficiency–voltage characteristic of OLEDs at each stage.

**Figure 9 micromachines-14-00278-f009:**
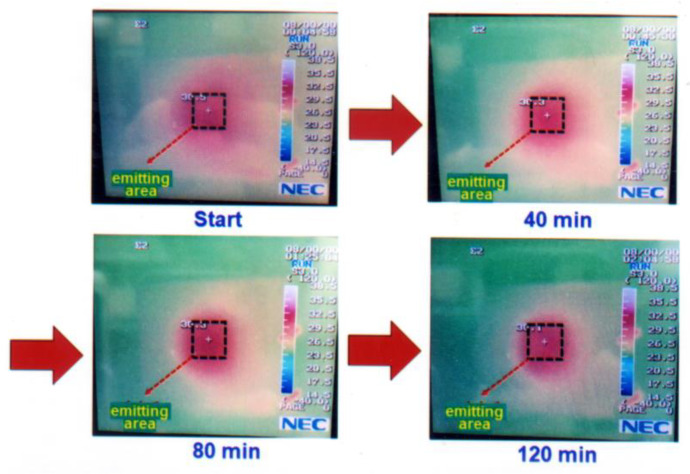
Under 5 mA constant current driven, the surface temperature and infrared thermal images of the OLED at different times (the middle square marked is the emitting area).

**Figure 10 micromachines-14-00278-f010:**
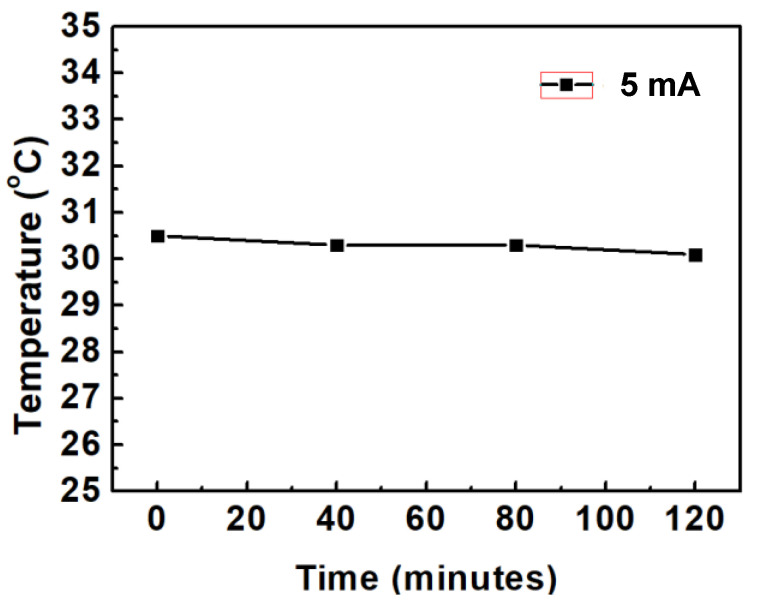
Under 5 mA constant current driven, the surface temperature of OLEDs vs measurement time.

**Figure 11 micromachines-14-00278-f011:**
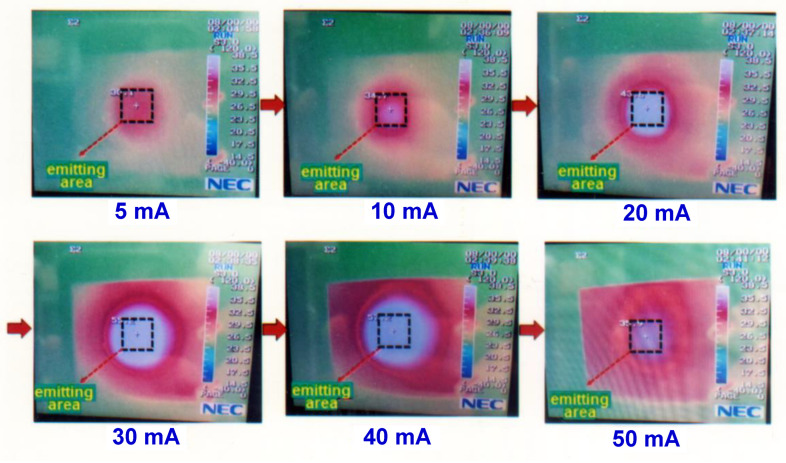
The variation of the OLED surface temperature with driving current.

**Figure 12 micromachines-14-00278-f012:**
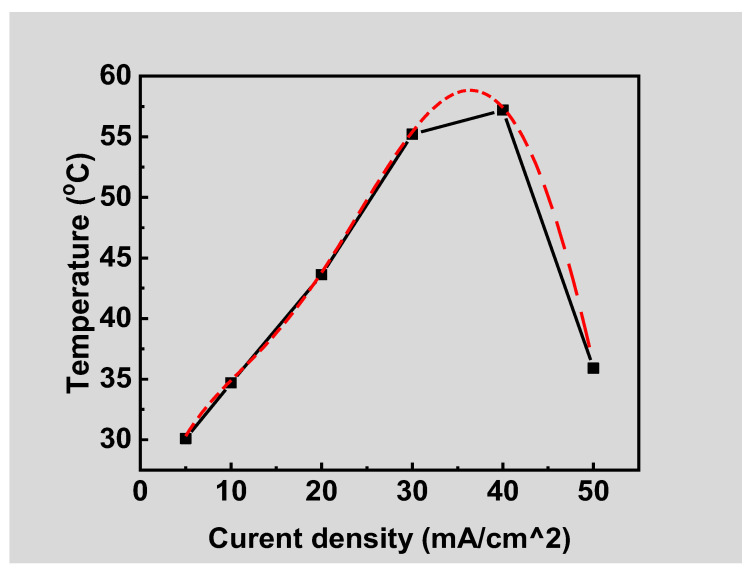
Relationship between OLED surface temperature and current density, where square dots are measured data and red dashed curve is numerical fitting.

**Table 1 micromachines-14-00278-t001:** Device structure parameters (unit: nm).

No	HIL	EML	HBL	ETL	EIL
	PEDOT:PSS (Spin × 1)	MADN: 13%UBD-07	TPBi (Evap.)	Alq_3_ (Evap.)	LiF (Evap.)
A	55	45 (spin × 1)	20	0	0.5
B		45 (spin × 1)	10	15	0.5
C		90 (spin × 2)			
D		90 (evap.)			
E	220 (spin × 4)	90 (evap.)	10	15	0.5

## Data Availability

Data are contained within the article.

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
