# Peer review of "Lifetime Comparisons of Organic Light-Emitting Diodes Fabricated by Solution and Evaporation Processes"

_micromachines, 2023, doi:10.3390/mi14020278_

Round 1
Reviewer 1 Report
1. Film morphology should be studied for the EML of devices E, F, and G should be given, for example, using the AFM technique.
2. The reason for the improved performance of device H relative to device G is still unclear, and the related explanation is not convincing enough due to a lack of solid evidence to support the authors' assumption.
The following explanation was given in the manuscript "After the PEDOT: PSS HIL thickness in device H was 203 increased from 55 to 220 nm, the number of holes injected into the EML was reduced, the 204 electron-hole current imbalance is reduced, and the luminance current efficiency is im-205 proved, as shown in Fig. 5." The authors suggested that the improved device performance is likely due to the enhanced charge balance in the device. Is there any experimental data to support it?
The comparison of current density versus applied voltage of hole-only and electron-only devices may be useful in determining device charge balance. And, the exciton quenching that occurs at the interface of PEDOT: PSS/EML has been reported in the literature. Is it possible that the increased PEDOT: PSS thickness could reduce such exciton quenching?
Author Response
Dear Editor
The revised manuscript entitled “Lifetime comparisons of organic light-emitting diodes fabricated by solution and evaporation processes” would like to be submitted to the Special Issue "Organic Light Emitting Diodes (OLEDs)" in the journal of Micromachine again. We received the invitation letter from Mr. Mars Tu and Miska Kuo. We have carefully revised the manuscript according to the reviewers’ comment. The revised parts are marked with red words in the revised manuscript. If there is any problem, please feel free to tell me.
Thank you very much for your kind consideration.
Sincerely yours,
Kuo Chun Huang

Reviewer 2 Report
In this work, the authors investigated the device lifetimes of various OLEDs according to the HTL thickness, ETL structure, and fabrication processes (solution, deposition). Also, they investigated the surface temperature change of the devices dependent on the operating current. The characterization was well performed, but many issues should be addressed before the consideration for recommendation.
1. Abstract should be summarized to show main result and discussion of the work.
2. The end of the introduction part should include the introduction and novelty of this work.
3. The chemical full names and molecular structures of all materials used in this work should be included.
4. The device number should be numbered as an order of A, B, C, D.. for easy understanding of the readers.
5. Since the Fig. 2 shows the fitting results, the description of fitting procedure should be addressed with describing Fig. 2.
6. Page 4. Line. 140, For the mobilities of TPBi and Alq3, the references or supporting data should be presented.
7. LUMO and HOMO levels should be expressed as minus values.
8. For all lifetime measurements, what is the measurement condition?
Author Response

(The authors gave the same response as above.)
